# Impact of *Camellia japonica* Bee Pollen Polyphenols on Hyperuricemia and Gut Microbiota in Potassium Oxonate-Induced Mice

**DOI:** 10.3390/nu13082665

**Published:** 2021-07-30

**Authors:** Yuanyuan Xu, Xirong Cao, Haoan Zhao, Erlin Yang, Yue Wang, Ni Cheng, Wei Cao

**Affiliations:** 1College of Food Science and Technology, Northwest University, 229 North TaiBai Road, Xi’an 710069, China; yy18435203605@163.com (Y.X.); haoan_zhao@126.com (H.Z.); 18235708524@163.com (E.Y.); wy15149677781@163.com (Y.W.); caowei@nwu.edu.cn (W.C.); 2College of Clinical Medicine, Jilin University, 828 XinMin Street, Changchun 130021, China; caoqr7016@mails.jlu.edu.cn; 3Bee Product Research Center of Shaanxi Province, Xi’an 710065, China

**Keywords:** *Camellia japonica* bee pollen polyphenols, hyperuricemia, kidney inflammation, TLR4/MyD88/NF-κB, NLRP3 inflammasome, gut microbiota

## Abstract

*Camellia japonica* bee pollen is one of the major types of bee pollen in China and exhibits antioxidant and anti-inflammatory activities. The aims of our study were to evaluate the effects and the possible mechanism of *Camellia japonica* bee pollen polyphenols on the treatment of hyperuricemia induced by potassium oxonate (PO). The results showed that *Camellia japonica* bee pollen ethyl acetate extract (CPE-E) owned abundant phenolic compounds and strong antioxidant capabilities. Administration with CPE-E for two weeks greatly reduced serum uric acid and improved renal function. It inhibited liver xanthine oxidase (XOD) activity and regulated the expression of urate transporter 1 (URAT1), glucose transporter 9 (GLUT9), organic anion transporter 1 (OAT1), organic cation transporter 1 (OCT1) and ATP-binding cassette superfamily gmember 2 (ABCG2) in kidneys. Moreover, CPE-E suppressed the activation of the toll-like receptor 4/myeloid differentiation factor 88/nuclear factor-κB (TLR4/MyD88/NF-κB) signaling pathway and nucleotide-binding oligomerization domain-like receptor family pyrin domain-containing 3 (NLRP3) inflammasome in PO-treated mice, and related inflammatory cytokines were reduced. CPE-E also modulated gut microbiota structure, showing that the abundance of *Lactobacillus* and *Clostridiaceae* increased in hyperuicemic mice. This study was conducted to explore the protective effect of CPE-E on hyperuricemia and provide new thoughts for the exploitation of *Camellia japonica* bee pollen.

## 1. Introduction

Hyperuricemia is a common metabolic disease with the obvious increasement of uric acid (UA) levels, which is linked with gout, disfunction of liver and kidney, systemic inflammation, cardiovascular disease and so on [1,2]. Recent research points to a rising incidence of hyperuricemia, which has become the second largest metabolic disease in China [3]. Almost a third of uric acid is from dietary purine catabolism, and the rest is produced endogenously. Excessive production of uric acid by the liver and insufficient excretion of uric acid by the kidneys can cause hyperuricemia, so the treatment of hyperuricemia is usually based on these two ways [4]. High levels of uric acid beyond the renal excretion capacity will precipitate and crystallize in the kidney, directly causing kidney injury. In addition, uric acid crystal deposition induces oxidative stress, activates the NLRP3 inflammasome and regulates the TLR4/MyD88/NF-κB signaling pathway, increasing the risk of kidney inflammation [5,6].

Gut microbiota refers to the whole set of microorganisms livings in the gut. There is increasing evidence that gut microbiota modulates immunity, the hormone system and host metabolism. Gut microbiota has been shown to be related to diabetes, obesity and ulcerative colitis [7,8]. In recent years, several studies supported the hypothesis that gut microbiota plays an important role in hyperuricemia [9,10]. Gut microbiota participates in the metabolism of purine and uric acid. Researchers found an altered gut microbiome in gout patients, with a significant decrease in gut bacteria expressing the gene of uricase [11]. Hyperuricemia affected the composition and metabolites of the gut microbiota, especially short chain fatty acid (SCFA). Guo et al. revealed the SCFA reduced in hyperuricemic mice [12]. However, few studies clarified the mechanism between the gut microbiota and hyperuricemia.

The general strategy for hyperuricemia focuses on decreasing uric acid using XOD inhibitors, such as allopurinol (AP) and febuxostat, and promoting excretion of uric acid using uricosuric agents, such as benzbromarone, probenecid and sulphinpyrazone [13,14]. However, the adverse reactions of XOD inhibitors and uricosuric agents, such as diarrhea, headaches, rashes, severe allergies, nephrotoxicity and so on, always limit their clinical utilization [15,16]. Therefore, finding more effective and non-toxic natural products against hyperuricemia is highly urgent. Polyphenols are the most common bioactive components of plants, which have many healthy functions such as anti-oxidation, anti-inflammatory, anti-tumor, anti-hypertension and so on [17,18]. A diet rich in polyphenols has been reported to have an anti-hyperuricemia effect manifesting as reduction of uric acid synthesis via XOD blocking, suppression of urate renal reabsorption and inhibition of uric acid secretion [19,20]. Parsley and celery are the most well-known antioxidant-rich herbs, Soliman et al. proved their anti-hyperuricemia activity at cellular and molecular levels [21]. Anthocyanin extract from purple sweet potato relieved hyperuricemia by weakening XOD activity [22].

Bee pollen, harvested as medicine and food for thousands of years, is rich in abundant nutriment, including proteins, polysaccharide, vitamins and polyphenols. Therefore, it has antioxidant, anti-inflammatory and immunoregulatory activities [23,24]. Lots of research on bee pollen was devoted to the physiological functions of polyphenols in recent years [25]. *Camellia japonica* bee pollen is one of the major categories of bee pollen in China. It is produced by honey bees by collecting pollen and mixing it with nectar and/or their secretion [26]. High contents of gallic acid (0.69–2.55 mg/g) were quantified from *Camellia japonica* bee pollen and possess strong antioxidant activities [27]. In our previous study, the polyphenols and biological activities of *Camellia japonica* bee pollen were studied, and the results showed that it possessed strong antioxidant activity and could prevent acute alcoholic liver injury [28]. It is worth noting that there are few reports on the prevention and treatment of hyperuricemia by *Camellia japonica* bee pollen polyphenols. Here, we firstly investigated the prevention of hyperuricemia and renal injury by *Camellia japonica* bee pollen polyphenols. We explored the antioxidant activity in vitro and the renal protection on hyperuricemic mice as well as the possible mechanism of CPE-E to provide theoretical references for the biological function of *Camellia japonica* bee pollen and its application on the treatment of hyperuricemia.

## 2. Materials and Methods

### 2.1. Camellia japonica Bee Pollen and Extraction (CPE)

*Camellia japonica* bee pollen was obtained from Hanzhong (Shaanxi, China). The pollen spores were examined under a microscope and identified as *Camellia japonica* bee pollen by comparing with the morphology of *Camellia japonica* bee pollen proposed by Kao et al. [27]. Bee pollen (2 kg) was ground into powder and refluxed three times with 75% ethanol for 2 h each time, then dissolved in methanol and water (1:1). The mixture was extracted with n-hexane and ethyl acetate [29]. Finally, it was concentrated at 45 °C under vacuum decompression to evaporate the solvent, and the ethyl acetate extract (CPE-E) (40.1 g) and n-hexane extract (CPE-H) (228 g) were obtained.

### 2.2. Chemicals and Materials

AP was obtained from Jiulian pharmaceutical Co. (Hefei, China). PO, Folin–Ciocalteu reagent, DPPH, Trolox, TPTZ, gallic acid, *p*-hydroxybenzoic acid, 2,4-dihydroxybenzoic acid, *p*-coumaric acid, rutin, ellagic acid, cinnamic acid, ferulic acid, quercetin, naringenin, Isoliquiritigenin and kaempferol were purchased from Sigma-Aldrich (Steinheim, Germany). Ethanol was purchased from Beijing Chemical (Beijing, China). Plasmid pBR322 DNA was obtained from Takara Biomedical (Dalian, China). Methanol was purchased from Merck (Darmstadt, Gemany). Other chemicals were purchased from Xi’an Chemical (Xi’an, China).

### 2.3. Phenolic Compound and Antioxidant Activity of CPE

#### 2.3.1. HPLC Analysis Determination of CPE

In order to identify the phenolic compound, Agilent 1100 separation module (Agilent Technologies, Waldbronn, Germany) was used and attached a diode-array detector with an SB-C18 column (250 × 4.6 mm, 5.0 μm). The mobile phase consisted of methanol (C) and 0.1% formic acid (D). We carried out the following procedure: C-D (15:85) at 0 min; C-D (17:83) at 5 min; C-D (30:70) at 10 min; C-D (40:60) at 25 min; C-D (50:50) at 35 min; C-D (55:45) at 40 min; C-D (60:40) at 55 min; C-D (65:35) at 65 min; C-D (70:30) at 70 min; C-D (70:30) at 75 min. During the whole process, column temperature was controlled at 30 °C [30]. The compound was detected at 280 nm. The content of the phenolic compound was calculated using calibration curves and was expressed as milligram per gram CPE.

#### 2.3.2. Determination of Total Phenolic Content (TPC)

A modified Folin–Ciocalteu approach, reported by Zhou, was employed [31] to determine TPC. CPE-E and CPE-H were prepared into 1 mg/mL of the sample solutions, respectively. One milliliter of sample was accurately blended with 1 mL of Folin–Ciocalteu reagent. Then, 5 mL of Na_2_CO_3_ solution (1 M) and 4 mL of distilled water were poured into the solution. After reacting for 1 h, the absorbance was read at 760 nm. TPC was expressed as gallic acid equivalents per gram CPE (mg GAE/g).

#### 2.3.3. Determination of Total Flavonoid Content (TFC)

A modified approach was used to quantify TFC [32]. One milliliter of sample at 1 mg/mL was blended with 0.4 mL of 5% NaNO_2_ solution. Naught point four milliliter of 10% Al(NO_3_)_3_ solution was pipetted 6 min later. After 6 min, four milliliters of 4% NaOH solution was added. Finally, the total was made up to 10 mL with 4.2 mL of 75% methanol and shaken well. The light absorption value was measured at 510 nm. TFC was expressed as rutin equivalents per gram CPE (mg RE/g).

#### 2.3.4. DPPH Radical Scavenging Activity

Scavenging DPPH radical capabilities of CPE were analyzed via a modified approach [33]. DPPH radical was prepared into 0.04 mg/mL methanol solution. Sample solution (0.4 mL) was pipetted to the brown plugged test tube, then, DPPH solution (5 mL) was blended and shaken well. The absorbance was measured at 517 nm after 1 h of light avoidance reaction. The following formula was applied to the calculation of DPPH clearance rate:$$\mathrm{Inhibition}\;\mathrm{of}\;\mathrm{DPPH}\;\mathrm{radical}\;\left(\%\right)=\frac{\left(A_{0}-A_{t}\right)}{A_{0}}\times 100$$
where $A_{0}$ represents absorbance of control and $A_{t}$ represents absorbance of the sample. The results were expressed as IC_50_ (the concentration of sample required to reach the inhibition of DPPH radical to 50%).

#### 2.3.5. Ferrous Ion-Chelating Activity

The approach of Singh was used to assess ferrous ion-chelating activity [34]. The reaction system contained 0.2 mL of sample (1 mg/mL), 0.1 mL of FeSO_4_ (1 mM), 0.3 mL of ferrozine (1 mM) and 4.4 mL of 75% methanol. After 10 min, the absorbance was measured at 562 nm. The results were expressed as Na_2_EDTA equivalents (mg Na_2_EDTA/g CPE).

#### 2.3.6. Ferric Reducing Antioxidant Power (FRAP)

In order to analyze FRAP, Mărghitaş’ approach was used [35]. FRAP solution was prepared with 25 mL of acetate buffer, 2.5 mL of TPTZ and 2.5 mL of FeCl_3_·6H_2_O solution. One milliliter of CPE at 1 mg/mL was blended with 4 mL of FRAP solution for 10 min in the dark. The absorbance was measured at 593 nm. The results were expressed as Trolox equivalents (mg Trolox/g CPE).

#### 2.3.7. Effect of CPE on DNA Oxidative Damage Induced by Hydroxyl Radicals

Effect of CPE was assessed in terms of the conversion of pBR322 plasmid DNA to open circular form [36]. Briefly, 1 µL of pBR322 DNA, 1 µL of FeSO_4_ (1 mM), 1 µL of 1% H_2_O_2_ and 4 µL of sample solution were added into the centrifuge tube, and the total volume was adjusted to 15 µL with 50 mM PBS. After incubating in a 37 °C water bath for 30 min, electrophoresis was performed and DNA band was detected. The result was quantified with Quantity One 4.6.2 software (Bio-Rad Company, California, USA).

### 2.4. Animals Experiment

#### 2.4.1. Hyperuricemic Mice Experiment Design

Male Kunming mice (18–22 g) obtained from Animal Center of Xi’an Jiaotong University (permission number: SCXK 2017-003). Ethical approval was given and adhered to the requirements of the Animal Ethics Committee of Northwest University. The mice were housed in a room with a 12 h light–dark cycle with a temperature of 22–25 °C and humidity of 40–60%. They were allowed free access to laboratory food and water. After a 7-day acclimation period, the mice were randomly grouped (each group = 10) into five groups, as follows: normal group (Group I), model group (Group II), CPE-E low dose group (Group III), CPE-E high dose group (Group IV), AP group (Group V). In the normal group and model group, the mice were given 0.5% CMC-Na solution by gavage. In the test groups, the mice were given 2 and 4 g of CPE-E/kg BW per day by gavage, respectively, and the mice of the AP group were given 5 mg/kg BW AP per day by gavage. All these administrations were conducted for 3 weeks [37]. On the 15th day, all mice except those in the normal group were administered intragastrically with 300 mg/kg BW of PO for the following 7 days.

After fasting overnight, blood was collected into a sterile centrifuge tube and centrifuged at 2500 rpm for 10 min. The mouse was sacrificed, and kidney and liver tissue was gathered. The kidney and liver were rinsed with 0.9% saline solution, then frozen with liquid nitrogen and stored in a refrigerator (−80 °C) until assayed. The organ index was calculated according to the following formulas:
$$\mathrm{Liver}\;\mathrm{index}\;\left(\%\right)=\frac{\mathrm{weight}\;\mathrm{of}\;\mathrm{liver}\;\left(g\right)}{\mathrm{body}\;\mathrm{weight}\;\left(g\right)}\times 100$$
$$\mathrm{Kidney}\;\mathrm{index}\;\left(\%\right)=\frac{\mathrm{weight}\;\mathrm{of}\;\mathrm{kidney}\;\left(g\right)}{\mathrm{body}\;\mathrm{weight}\;\left(g\right)}\times 100$$

#### 2.4.2. Serum, Liver and Kidney Biochemical Analysis

Serum was collected by centrifugation from blood. Parts of kidney and liver were homogenized with cold physiological saline and 10% solution, respectively. Serum UA, creatinine (Cr), blood urea nitrogen (BUN), hepatic XOD, superoxide dismutase (SOD), glutathione (GSH) and malondialdehyde (MDA) were tested using commercial kits (Jiancheng Biotech, Nanjing, China). Serum and renal inflammatory factors TNF-α, IL-1β, IL-6 and IL-18 were characterized by immunoassay using ELISA kits produced by Fusheng Industrial (Shanghai, China).

#### 2.4.3. Histopathological Examination

A portion of kidney tissue was fixed in 4% paraformaldehyde solution, then dehydrated and embedded in paraffin under the guidance of experimental protocol. Heamatoxylin and Eosin (H&E) were used for staining, the slides were cleared and examined under light microscopy at 200× magnification. Renal histological damage was evaluated by a semiquantitative score [38]. The blinded histopathological examination of the kidney tissue was performed by multiple investigators. The method was based on glomerulopathy (0–4, from normal glomerular structure to most glomerular atrophy), tubulopathy (0–4, from normal structure of tubules to most tubules) and renal interstitial inflammatory infiltration (0–4, from non-inflammatory cells to a large number of inflammatory cells).

#### 2.4.4. Quantitative Real-Time PCR Analysis

A Takara minibest RNA extraction kit (Takara, Japan) was used to extract RNA from the kidneys. RNA was reverse-transcribed using a Primescript RT master mix kit (Takara, Japan). The cDNA samples were amplified in duplicate. Quantitative real-time PCR (Q-PCR) was carried out on the Gentier 96E (Tianlong Technology, Xi’an, China) with TB Green Premix Ex Taq. Q-PCR was run under the following conditions: 95 °C, 30 s; 95 °C, 5 s; 60 °C, 30 s. The primer sequences of genes such as URAT1, GLUT9, OAT1, OCT1, ABCG2, TLR4, MYD88, NF-κB, NLRP3, apoptosis-associated speck-like protein containing a CARD (ASC) and Caspase-1 are listed at Appendix A. Gene expression was normalized with β-actin, and 2^−^^ΔΔCt^ was used to calculate the result.

#### 2.4.5. Gut Microbiota Analysis

Samples of colonic contents (Z, M, D, H and Y represent Group I, Group II, Group III, Group IV and Group V, respectively; n = 5/group) were stored in a refrigerator (−80 °C). DNA was extracted with Fast SPIN extraction kits (MP Biomedical, Santa Ana, CA, USA). In order to assess the quality of the extracted DNA, agarose gel electrophoresis was performed. The V3-V4 region of bacterial 16S rRNA gene was amplified. AgencourtAMPure Beads (Beckman Coulter, Indianapolis, IN, USA) was used to purify amplicons, followed by quantification using the PicoGreendsDNA Assay Kit (Invitrogen, Carlabad, CA, USA). Paired-end sequencing was run in an Illumina MiSeq instrument with a MiSeq Reagent Kit v3 (Shanghai, China). Then, all reads were scored for quality (QIIME2, v1.8.0), and poor-quality and short reads were removed. The data was analyzed according to the method of Callahan with microbial ecology platform (QIIME2, v1.8.0) [39]. Operational taxonomy units (OTUs) with similarity at 97% were selected to calculate the diversity index. Principal coordinates analysis (PCoA) examined the abundance and diversity of OTUs.

#### 2.4.6. SCFA Analysis

The content of SCFA was determined by Agilent 5977B GC-MS with a DB-FFAP capillary column (Agilent, Palo Alto, USA). The running procedure was as follows: the initial temperature was 80 °C and at a rate of 10 °C/min, raised to 180 °C (kept 6 min), then to 200 °C at a rate of 1 °C/min (kept 14 min), finally, up to 230 °C at a rate of 4 °C/min (kept 30.5 min). In the whole process, injector temperature remained constant at 270 °C. The sample injection volume was 1 μL. Freeze-dried fecal samples were used to determine SCFA. Sample preparation: 0.2 g of feces and 5-times distilled water were placed in a beaker and sonicated, then centrifugated at 10,000 rpm for 5 min, and the supernatant was sucked through an aqueous membrane. One hundred microliters of 50% H_2_SO_4_ and 1 mL of diethyl ether solution was added, shaken and centrifuged at 10,000 rpm for 5 min. After 30 min, the supernatant was taken and stored in the sample bottle.

### 2.5. Statistical Analysis

All the tests were performed in triplicate. Data were analyzed by one-way ANOVA method in SAS followed by Tukey’s test (SAS, NC, USA) and shown as the mean ± standard deviation. *p* < 0.05 was considered to be statistically significant.

Integration of the original sequencing data, classification of OTUs, alpha-diversity analysis (including Chao1, Observed species, Simpson and Shannon), beta-diversity (including PCoA and hierarchical clustering) and microbial composition analysis were implemented with QIIME (v1.8.0).

## 3. Results and Discussion

### 3.1. The Phenolic Compound and Antioxidant Activity of CPE-E and CPE-H

The chromatograms of CPE were presented in Figure 1, CPE-E showed six phenolic compounds: gallic acid, *p*-hydroxybenzoic acid, ferulic acid, ellagic acid, quercetin and kaempferol. Quercetin and kaempferol were the abundant compounds in the quantified phenolic compounds, reaching 6.85 and 7.29 mg/g (Table 1), respectively. Three phenolic compounds were identified in CPE-H, including *p*-hydroxybenzoic acid, ferulic acid and ellagic acid. Ellagic acid of CPE-H was two times higher than that of CPE-E. We determined the TPC, TFC and antioxidant activity of CPE-E and CPE-H, and the results are shown in Table 2. TPC and TFC of CPE-E were higher than those of CPE-H (*p* < 0.05).

Table 2 shows the results of DPPH scavenging activity, FRAP and ferrous ion-chelating ability of CPE-E and CPE-H. The IC_50_ value of CPE-E was only 0.20 mg/mL, whereas 5.03 mg/mL was obtained in CPE-H, demonstrating that CPE-E possessed stronger free radical-scavenging capability than CPE-H. FRAP of CPE-E was 0.49 mg Trolox/mg and significantly higher than that of CPE-H (*p* < 0.05). Ferrous ion-chelating activity of CPE-E was 29.10 mg Na_2_EDTA/g, remarkably higher than that of CPE-H (*p* < 0.05). From Figure 2, two extracts of *Camellia japonica* bee pollen had a protective effect on plasmid DNA oxidative damage, and the effect of CPE-E was significantly better than that of CPE-H. When the concentration of CPE-E increased from 0.70 to 2.70 μg/mL, the proportion of supercoiled DNA increased from 66.85% to 97.65%, which had a dose-response relationship.

In short, CPE-E had better antioxidant activities in in vitro study. Therefore, CPE-E was selected for evaluation of the protective effect on hyperuricemia.

### 3.2. Effect of CPE-E on Body and Organ Weight and Serum Biochemical Parameters in Mice

Organ indexes can be a significant indicator of visceral health. The result of body weight, liver weight, kidney weight and organ indexes are presented in Figure 3A–C, which showed no significant differences, indicating that administration of CPE-E did not cause any damage to the organ.

The level of serum UA is an important index of the success of a hyperuricemic model. Figure 3D shows that the UA levels of Group II were 2.54 times higher than that in Group I (*p* < 0.05), which indicated that the model of hyperuricemia in this experiment was successfully established. Supplement with CPE-E (2, 4 g/kg BW) and AP (5 mg/kg BW) for three weeks remarkably decreased UA values by 53.45%, 72.83% and 75.38%, respectively, compared to Group II. The protective effect of 5 mg/kg BW AP and 4 g/kg BW CPE-E was similar.

Serum Cr and BUN levels are key clinical indicators of renal function. As shown in Figure 3E, Cr and BUN values of Group II showed an evident increase compared to those in Group I. Pretreatment with 2, 4 g/kg BW of CPE-E and AP decreased Cr levels by 32.32%, 46.14% and 44.71% (*p* < 0.05), and decreased the BUN levels by 12.21%, 23.93% and 25.61% (*p* < 0.05), respectively. This suggested that CPE-E could effectively improve kidney function.

### 3.3. Effect of CPE-E on XOD Activity in Liver

XOD is a flavoprotein which catalyzes oxidation of hypoxanthine and xanthine to uric acid [40]. Therefore, inhibition of XOD activity has turneda into an efficacious way for holding back the production of uric acid. PO administration significantly increased liver XOD activity in Group II, which was 33.90% higher than that in Group I (Figure 3F). Pretreatment with 2 and 4 g/kg BW of CPE-E significantly attenuated liver XOD activity. Moreover, there was no significant difference among normal mice, CPE-E and AP mice, indicating that CPE-E could absolutely inhibit the increase of liver XOD activity induced by PO.

### 3.4. Effect of CPE-E on Antioxidant Status

SOD, GSH and MDA play pivotal roles during oxidative stress. When PO-induced oxidative stress developed, MDA increased, accompanied by decrease of SOD and GSH, as shown in the results of Group II (Figure 3G–I). Administrating 2 and 4 g/kg BW of CPE-E showed a significant decrease in liver MDA contents and increase in liver SOD activities (*p* < 0.05). Additionally, 4 g/kg BW of CPE-E significantly increased liver GSH contents. These results suggested that the application of CPE-E could weaken the oxidative stress caused by PO.

### 3.5. Effect of CPE-E on the Related Transporters in Renal Tissue

The mRNA expression of transporters URAT1, GLUT9, OAT1, OCT1 and ABCG2 was analyzed by Q-PCR to evaluate urate excretion in renal tissue. Figure 4A exhibits the results. After induction by PO, the expression of mURAT1 and mGLUT9 in Group II increased 1.51 times and 0.83 times compared to that in Group I (*p* < 0.05), respectively. CPE-E at 2 and 4 g/kg BW markedly downregulated their expression compared to that in Group II. The expression of mOAT1, mOCT1 and mABCG2 was clearly downregulated in Group II. Treatments with 4 g/kg BW of CPE-E and 5 mg/kg BW of AP were found to upregulate the mRNA levels of OAT1, OCT1 and ABCG2 (*p* < 0.05). However, 2 g/kg BW of CPE-E had no significant impact, indicating that only 4 g/kg BW of CPE-E could have a remarkable effect on urate transport.

### 3.6. Histopathological Analysis

Renal inflammation is a pathological feature of hyperuricemia in clinical settings. The results of renal histopathological analysis are displayed in Figure 5. Mice in Group I maintained normal kidney structure, while the morphology in Group II was significantly changed, with renal interstitial infiltrating inflammatory cells, partial atrophic glomerulus and renal tubules dilating. Administration with different concentrations of CPE-E, renal damage was alleviated by reducing the infiltration of inflammatory cells in renal interstitium and weakening renal tubular expansion and glomerular atrophy. Morphology of renal tissue analysis combined with histopathological score indicated that CPE-E could ameliorate the kidney damage caused by hyperuricemia. More importantly, AP group showed that brush border of renal tubular epithelial cells disappeared, and renal tubules dilated. Surprisingly, none of this was observed in CPE-E group. These suggested that AP had side effects in the kidneys, but CPE-E did not. Therefore, CPE-E can be well-used to treat hyperuicemia.

### 3.7. Effect of CPE-E on Inflammatory Cytokines in Renal and Serum

In this study, renal and serum IL-6, TNF-α, IL-1β and IL-18 were determined. As shown in Figure 6, treatment with 300 mg/kg BW of PO could markedly increase the levels of IL-6, TNF-α, IL-1β and IL-18 both in the kidneys and serum (*p* < 0.05). Surprisingly, 4 g/kg BW of CPE-E and AP notably reduced the contents of these inflammatory biomarkers compared to those in Group II.

### 3.8. Effect of CPE-E on TLR4/MyD88/NF-κB and NLRP3/ASC/Caspase-1 Signaling Pathways in Renal

TLR4/MyD88/NF-κB and NLRP3/ASC/Caspase-1 are primary signaling pathways closely related to inflammation which mediate inflammatory factors IL-6, TNF-α, IL-1β and IL-18. As shown in Figure 4B, the transcription of TLR4, MyD88, NF-κB, NLRP3, ASC and Caspase-1 were significantly upregulated by PO treatment. However, both 4 g/kg BW of CPE-E and 5 mg/kg BW of AP treatment decreased the transcription of all genes (*p* < 0.05). Finally, 2 g/kg BW of CPE-E could inhibit the expression of Caspase-1.

### 3.9. CPE-E Alter Gut Microbiota

To study the impact of CPE-E on gut microbiota, 16S rRNA sequencing technology was performed. We produced a total of 1,864,704 sequences, with an average of 69,041 sequences in Group I, 67,659 sequences in Group II, 63,090 sequences in Group III, 68,054 sequences in Group IV and 63,739 sequences in Group V. Of the sequences, 77.73% was 420–440 base pairs (Appendix A). With the increase of the total number of sequences in current samples, rarefaction curve gradually became flat, which showed that the sequencing results were adequate to reflect the diversity of samples (Appendix A). Venn diagram presents the number of OTUs common and unique to each group. In Figure 7A, the number of OTUs shared by Group II (M) and Group I (Z) was 966, while the number of OTUs shared by Group III (D) and Group I (Z), Group IV (H) and Group I (Z), and Group V (Y) and Group I (Z) increased to 1082, 1199 and 1317, respectively.

Alpha-diversity analysis showed the diversity of each sample community. Among them, Chao1 and Observed species indexes reflect community richness, while Shannon and Simpson indexes reflect community richness and evenness. Compared with other groups, the diversity of the PO-induced model group was reduced in Simpson and Shannon indexes. Surprisingly, Simpson and Shannon indexes enhanced significantly in Group III and Group IV (*p* < 0.05) to converge with those indexes of Group I (Table 3). These indicated that CPE-E at 2 and 4 g/kg BW had an effect on the diversity of the gut microbiota, and although species richness increased but was not significant (Chao1 and Observed species), species evenness and relative abundance changed significantly (Simpson and Shannon). The rank abundance curve showed that species richness in Group III and Group IV was higher than that in Group II (Appendix A). These results indicated that CPE-E played a prominent role in diversity of gut microbiota.

PCoA examine the similarity of gut microbial structure. As shown in Figure 7B, the sample of Group II clustered together and away from Group I. Group III and Group IV were farther apart and showed a unique gut microbiota structure compared to that of Group II. This result demonstrated that CPE-E could induce a gut microbial structure which more closely resembles that in a healthy gut, and a high dose of CPE-E gave more obvious changes. The similarity between different samples could also be evaluated by hierarchical clustering. The similarity of Group I and Group III, Group IV was higher than that of Group I and Group II. The community structure and abundance of bacteria in Group I, Group III, Group IV and Group V were similar, mainly composed by *Firmicutes* and *Bacteroidetes* (Figure 7C). In Group II, *Firmicutes* reduced significantly and *Bacteroidetes* increased significantly, and the change was reversed after gavaging CPE-E, indicating that CPE-E could influence the species composition of the gut microbiota.

Figure 7D showed the relative abundance of the primary microbiota component at the phylum level. The top four phyla were *Firmicutes*, *Bacteroidetes*, *Actinobacteria* and *Proteobacteria*, which accounted for close to 96% of the total (Appendix A). The significant increase in *Firmicutes* and reduction in *Bacteroidetes*, *Actinobacteria* and *Proteobacteria* were all reversed by CPE-E treatment, whereas the abundance of *Actinobacteria* was only restored with 4 g/kg BW of CPE-E. At the genus level, the PO group had higher *Staphylococcaceae* and lower *Lactobacillus*, *Allobaculum* and *Adlercreutzia* (*p* < 0.05) compared with those in the control group, while in the CPE-E and AP group, *Lactobacillus* and *Staphylococcaceae* were improved to varying degrees (Figure 7E). The beneficial bacteria *Lactobacillus* in 4 g/kg BW of CPE-E group increased 66.26% compared to that in the PO group. *Allobaculum* and *Adlercreutzia* were restored only in the 2 g/kg BW of CPE-E group. As shown in Figure 7F, *Bacilli*, *Bacteroidia* and *Clostridia* were the most abundant gut microbiota in all treatments at the class level. CPE-E and AP treatment reduced *Bacteroidia* and increased *Clostridia*. *Clostridia* in CPE-E and AP groups was similar to that in the control group. *Bacilli* was restored only in the group of 4 g/kg BW CPE-E.

Heatmap reflects the differences in microbial composition. In the phylum level, we analyzed the 20 most-abundant bacteria. CPE-E partly reduced some bacterial species which were increased by PO induction, consisting of *Firmicutes*, *Nitrospirae*, *Gemmatimonadetes* and *Cyanobacteria*. In Group II, the abundance of *Proteobacteria*, *Tenericutes*, *Actinobacteria* and *TM7* decreased after the intake of CPE-E improved the situation and the high-dose of CPE-E had a better effect (Figure 7G). The absolute abundance of gut microbiota based on the phylum level was shown in Figure 7H. *Proteobacteria*, *Actinobacteria* and *TM7* decreased significantly in the PO-induced model group and effectively improved in the 4 g/kg BW of CPE-E group. Additionally, CPE-E at 4 g/kg BW alleviated the increase of *Firmicutes.*

### 3.10. Effect of CPE-E on the Concentrations of SCFA

The concentrations of four SCFAs were measured (Figure 8). Compared to those in Group I, acetic acid, propionic acid and butyric acid contents in Group II were significantly decreased by 61.5%, 62.2% and 66.7%, respectively. After administration of 4 g/kg BW of CPE-E, the contents of acetic acid and butyric acid increased in Group IV (*p* < 0.05). In addition, there were no significant differences in concentration of valeric acid among the five groups.

## 4. Discussion

Uric acid is the end product of purine metabolism, most of which is degraded into allantoin through the action of uricase. The production and excretion of uric acid in normal humans basically maintain a dynamic balance, but when blood uric acid is produced too strongly and/or secreted too little, it will cause hyperuricemia [41]. Blood uric acid content in the human body is one of the main indicators for the diagnosis of hyperuricemia. PO is an uricase inhibitor that can specifically block the degradation of uric acid to allantoin, thereby increasing blood uric acid. It has been widely used in formation of hyperuricemia in mice [42]. In this study, treatment with PO significantly increased serum uric acid compared to that in the control group, which suggested that a hyperuricemic mice model was successfully established. Administration of 2 and 4 g/kg BW of CPE-E significantly reduced the serum uric acid levels, revealing that *Camellia japonica* bee pollen had an anti-hyperuricemia effect.

XOD is the key factor of purine catabolism in the human body which catalyzes the metabolisms of hypoxanthine and xanthine to uric acid [43]. Inhibition of XOD activity is beneficial to reducing uric acid. In our study, PO treatment enhanced liver XOD activity. CPE-E significantly inhibited XOD activity, contributing to reducing the production of uric acid. Based on this, we deduced that inhibition of XOD activity may be one of the mechanisms of *Camellia japonica* bee pollen polyphenols against hyperuricemia. Lin et al. proved that *Rhizoma Alpiniae Officinarum* extract has a significant hypouricemic effect through inhibiting XOD activity [44].

Many studies have confirmed that hyperuricemia is not only due to excessive production of uric acid, but also insufficient excretion. Uric acid exists in the form of free urate and is excreted through the kidney, including glomerular filtration, renal tubular reabsorption, renal tubular re-secretion and post-secretion reabsorption [45]. Uric acid transporter proteins in the kidneys are involved in the pathogenesis of hyperuricemia. URAT1 and GLUT9 specifically mediate urate reabsorption in proximal renal tubules, which are considered to be the most promising therapeutic targets for hyperuricemia. ABCG2 locates on the membranous side of the proximal convoluted tubule and is reported to secrete the urate transporter [46,47]. Administration of a high dose of CPE-E downregulated kidney URAT1 and GLUT9 mRNA levels and up-regulated ABCG2 mRNA levels, which reduced the reabsorption and increased the excretion of uric acid. OAT1 participates in uric acid excretion at the basolateral side of proximal renal tubules and plays an important role in the progress of chronic kidney diseases. OCT1 is known to be related to normal renal function because it mediates the excretion of organic cations in proximal tubules and thus maintains the renal organic ion balance [48]. In our study, CPE-E increased the level of mOAT1 and mOCT1 in the kidneys of hyperuricemic mice. These organic transporters, which play critical roles on excreting uric acid, may be one of the important targets of *Camellia japonica* bee pollen in the treatment of hyperuricemia.

Usually, hyperuricemia is accompanied by oxidative stress because a low level of uric acid is an important endogenous antioxidant, which can chelate metals and scavenge oxygen-free radicals in vivo [49]. However, high levels of uric acid can promote oxidation, which will disrupt the body’s redox balance and produce large amounts of reactive oxygen species (ROS), thereby inducing the body’s oxidative stress [50]. In the present study, we confirmed the antioxidant and anti-inflammatory effects of CPE-E on hyperuricemic mice. CPE-E was rich in quercetin, kaempferol and gallic acid. It had a TPC of 136.63 mg GAE/g and TFC of 67.49 mg RE/g, which was higher than that of 13 bee pollens from Turkey [51]. Usually, the antioxidant components that comprise phenolic acids and flavonoids are the contributors to antioxidant capability. The results of DPPH radical scavenging capability, FRAP, ferrous ion-chelating ability and protective effect on DNA oxidative damage showed that CPE-E had super antioxidant capacity in vitro. The phenolic compound in CPE-E owned hydroxyls and carbonyl groups in the benzene ring, acting as electron-donating substituents or hydrogens that could reduce free radicals and form stable phenoxy groups [52]. Therefore, they could be used as reductants, hydrogen donors and singlet oxygen quenchers to show the antioxidant activities.

The antioxidant system of the human body is complex, and antioxidants can be synthesized by themselves. SOD catalyzes the removal of superoxide anions, produces hydrogen peroxide and scavenges lipid peroxide. GSH also participates in a regulatory response to decrease oxidative stress by metabolizing lipid hydroperoxide. MDA is the end-product of lipid peroxide and can reflect the degree of liver injury [53]. The liver is the main organ affected by oxidative stress [54]. In our study, CPE-E increased antioxidant biomarkers SOD activities and GSH contents and decreased MDA contents. These results confirmed that *Camellia japonica* bee pollen polyphenols had antioxidant activity in vivo, ameliorating these biomarkers related to oxidative stress. In the previous study, dietary honey polyphenols improved serum antioxidant ability in mice [55]. Inhibition of oxidative stress provided us with another explanation for anti-hyperuricemia activity. The higher uric acid level is prone to developing oxidative stress, which further stimulates renal inflammation and damage. Yu et al. confirmed that uric acid directly linked to oxidative stress which could lead to other diseases [56].

Inflammation is a pathological feature of hyperuricemia. Uric acid stimulates inflammatory mediators and causes inflammatory reaction. Hyperuricemia can promote the expression of inflammatory cytokines, such as IL-1β, TNF-α, and IL-6 [57]. The present study showed that CPE-E significantly ameliorated IL-6, TNF-α, IL-1β and IL-18 in renal tissue and serum of hyperuricemic mice, which was consistent with a previous study in which Chen et al. found that curcumin has a beneficial effect on hyperuricemia by inhibiting inflammatory cytokines IL-1β and IL-18 [58]. What is more, renal histological assessment revealed that CPE-E could attenuate renal injuries, which may be attributed to the regulation of inflammatory cytokines. Toll-like receptors (TLRs) are known to cause inflammation by triggering the innate immune system. TLR4 acts as a promoter in the inflammatory response chain and regulates many inflammatory cytokines in hyperuricemia. Studies have reported that TLR4/NF-κB is a key transcriptional pathway regulating the secretion of inflammatory cytokines (such as IL-6, TNF-α and IL-1β). NF-κB is activated rapidly via the MyD88-dependent pathway, which increases the transcription of downstream inflammation-related genes [59]. NLRP3 inflammasome is a member of the nucleotide oligomerization domain (NOD)-like receptor family, which promotes innate immune defense through the maturation of inflammatory cytokines (such as IL-1β and IL-18). NLRP3 interacts with ASC, activates Caspase-1 and produces inflammation [5]. Uric acid increases the TLR-induced inflammatory factor and amplifies human gut cell inflammation through the activation of the TLR4-NLRP3 signaling pathway [60]. Han et al. alleviated hyperuricemia by suppressing the TLR4/MYD88/NF-κB signaling pathway and NLRP3 inflammasome [57]. In this study, CPE-E inhibited the TLR4/MYD88/NF-κB signaling pathway and decreased the expression of NLRP3/ASC/Caspase-1, downregulated the transcription of IL-6, TNF-α, IL-1β and IL-18, thereby relieving renal inflammation. This may be contributed to the anti-oxidation characteristic of *Camellia*
*japonica* bee pollen polyphenols. A similar trend was observed by An et al. in a study in which Isoorientin extracts treatment regulated the TLR4-NLRP3 inflammasome signaling pathway. Therefore, inhibiting the TLR4 and NLRP3 pathways effectively reduced the release of inflammatory cytokines and at the same time attenuated the renal damage of hyperuricemia.

Clinical reports have shown that kidney damage is characterized by high serum Cr and BUN levels [61]. CPE-E significantly reversed the increase of Cr and BUN in hyperuricemic mice. Administration of 4 g/kg BW of CPE-E provided the strongest recovery ability to renal damage. Histopathological examination also revealed that high-dose CPE-E treatment markedly alleviated renal damage. Although AP clearly lowered Cr and BUN levels, its long-term intake was proven to cause severe hypersensitivity, granulocytosis and aggravate renal toxicity [62,63]. These results suggested that *Camellia japonica* bee pollen polyphenols significantly recovered renal function in mice.

Polyphenols are powerful antioxidants that protect our bodies from diseases such as colitis, cancer, diabetes and cardiovascular disease. Studies have shown that polyphenols have protective effects on health by inhibiting these diseases [64,65]. In our study, CPE-E rich in phenolic compounds prevent hyperuricemia by reducing uric acid, attenuating oxidative stress, regulating uric acid transporter proteins and suppressing inflammation in mice. The dosage 4 g/kg in mice, equivalent to human dosage of 0.32 g/kg, could be used in clinical treatment of hyperuricemia.

Alterations in the composition of the gut microbiota and its metabolites in response to CPE-E may be another key factor contributing to the alleviation of hyperuricemia and renal inflammation. The gut microbiota is a group of microorganisms present in the colon, which helps maintain human health. There are more than 5000 kinds of bacteria in the human intestine, most of which are strictly anaerobic bacteria, mainly composed of two dominant groups: *Bacteroidetes* and *Firmicutes*, followed by *Proteobacteria*, *Fusobacteria* and *Actinobacteria* [66]. A potential role of the gut microbiota in metabolic diseases has recently been revealed [67]. Normal mice that received gut microbiota transplants from hyperuricemic mice further increase uric acid [68]. In the present study, the abundance of *Firmicutes* increased significantly in hyperuricemic mice, and the abundance of *Bacteroidetes*, *Proteobacteria* and *Clostridium* decreased significantly, which is consistent with the results of Wan et al. [59]. After CPE-E (4 g/kg BW) treatment, it recovered to the level close to the that in the normal group. *Lactobacillus* is a probiotic bacteria whose abundance in the gut directly influences host health. It has been studied to create a close relationship with polyphenols. In addition, *Lactobacillus* has an anti-hyperuricemia effect, synthesize uricase and, finally, degrade uric acid into urea [69]. CPE-E treatment increased the abundance of *Lactobacillus*. *Clostridiaceae* have the capability to degrade uric acid. Xing et al. sequenced the gut microbiota of 90 patients with gout and found that *Clostridium* in patients was significantly reduced [70]. *Clostridium*, *Ruminococcus* and *Bifidobacterium* are known to produce SCFA. Studies have confirmed SCFA, particularly propionic acid and butyric acid provide energy for excretion of uric acid by the cells in the intestinal wall [71]. Supplementation with SCFA can significantly improve renal function [72]. In this study, the concentrations of acetic acid and butyric acid were significantly increased after gavage of high dose CPE-E. These alterations in gut microbiota were proposed to be one of mechanisms of CPE-E treatment to relieve hyperuricemia. However, whether the beneficial effect of CPE-E is mediated by the gut microbiota should be further explored by fecal microbiota transplantation experiments.

## 5. Conclusions

In conclusion, CPE-E possessed abundant polyphenols and high antioxidant capabilities. It cut down production of uric acid and promoted excretion of uric acid in hyperuricenic mice. Moreover, CPE-E improved antioxidant status and decreased inflammation in PO-induced hyperuricemic mice to alleviate renal damage, and possible mechanisms included inhibition of the TLR4/MyD88/NF-κB pathway and suppression of the NLRP3 inflammasome. In addition, CPE-E altered the gut microbiota structure and increased the abundance of beneficial bacteria such as *Lactobacillus* and *Clostridium*, and the content of short-chain fatty acids also increased accordingly. Whether or not the anti-hyperuricemia and anti-inflammatory effects of polyphenols are mediated by the gut microbiota will require future exploration. This study will provide an effective treatment for hyperuricemia and put forward a new application of *Camellia japonica* bee pollen.

## Figures and Tables

**Figure 1 nutrients-13-02665-f001:**
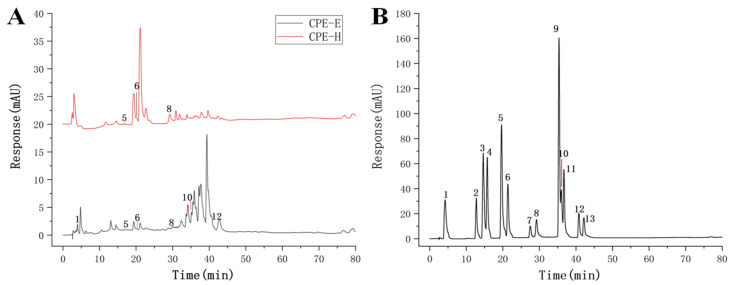
Chromatogram of the identified phenolic compound. Chromatogram of CPE (**A**), Chromatogram of standard (**B**).

**Figure 2 nutrients-13-02665-f002:**
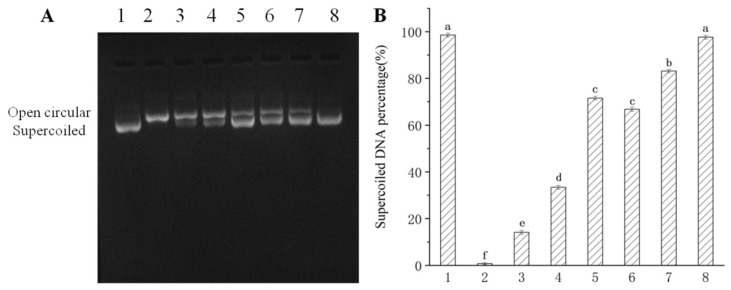
Protective effect of *Camellia japonica* bee pollen polyphenols on hydroxyl radical-mediated pBR322 DNA damage. Electrophoregram of DNA (**A**), supercoiled DNA percentage (**B**). Lane 1, 1 μL pBR 322 DNA; Lane 2, 1 μL pBR 322 DNA + 1 μL of 1% H_2_O_2_ + 1 μL of 1 mM FeSO_4_; Lane 3–8, 1 μL pBR 322 DNA + 1 μL of 1% H_2_O_2_ + 1 μL of 1 mM FeSO_4_ + 0.7, 1.3, 2.7 μg/mL of CPE-H and CPE-E, respectively. Different lowercase letters represent significant differences (*p* < 0.05).

**Figure 3 nutrients-13-02665-f003:**
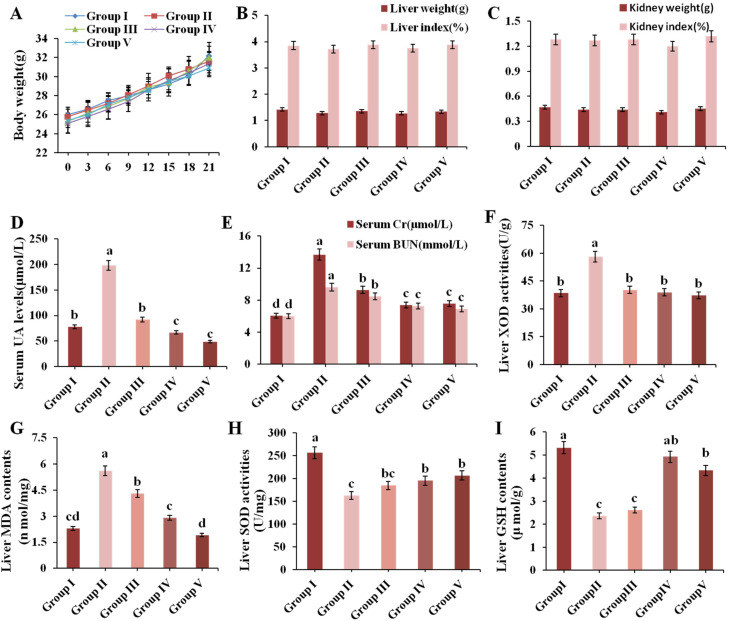
Effect of CPE-E on body weight (**A**), liver weight and index (**B**), kidney weight and index (**C**), serum UA levels (**D**), serum Cr and BUN levels (**E**), liver XOD activities (**F**), MDA contents (**G**), SOD activities (**H**) and GSH contents (**I**) in mice. Group I was the control group. Group II was given potassium oxonate. Group III and Group IV were given potassium oxonate plus 2 and 4 g/kg BW of CPE-E, respectively. Group V was given potassium oxonate plus 5 mg/kg BW of allopurinol. Different lower case letters correspond to significant differences at *p* < 0.05.

**Figure 4 nutrients-13-02665-f004:**
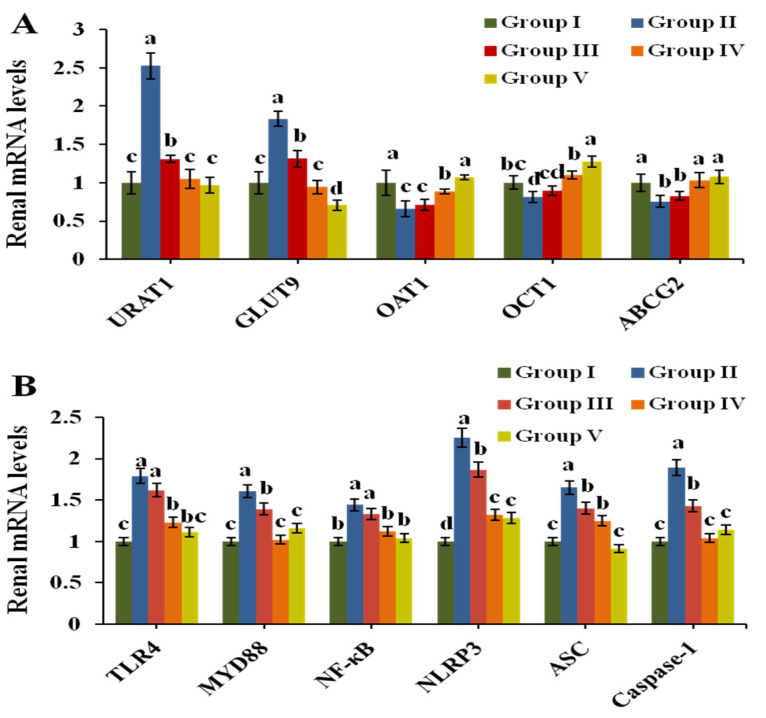
Effect of CPE-E on mRNA levels of renal URAT1, GLUT9, OAT1, OCT1 and ABCG2 (**A**), TLR4, MyD88, NF-κB, NLRP3, ASC and Caspase-1 (**B**) in hyperuricemic mice. Group I was control group. Group II was given potassium oxonate. Group III and Group IV were given potassium oxonate plus 2 or 4 g/kg BW of CPE-E, respectively. Group V was given potassium oxonate plus 5 mg/kg BW of allopurinol. Different lower case letters correspond to significant differences at *p* < 0.05.

**Figure 5 nutrients-13-02665-f005:**
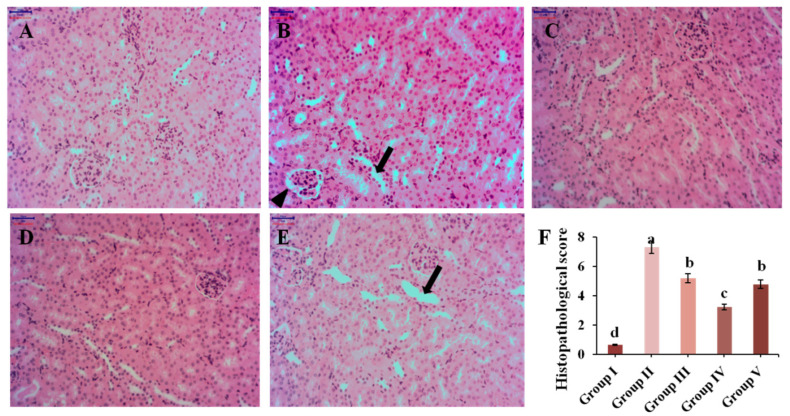
The effect of CPE-E on pathological injury in the kidneys (×200 H&E). Group I was the control group (**A**), Group II was given potassium oxonate (**B**), Group III was given potassium oxonate plus 2 g/kg BW of CPE-E (**C**), Group IV was given potassium oxonate plus 4 g/kg BW of CPE-E (**D**), Group V was given potassium oxonate plus 5 mg/kg BW of allopurinol (**E**), the histopathological score of mice (**F**). (➔, renal tubular expansion; ▲, glomerular atrophy). Different lower case letters correspond to significant differences at *p* < 0.05.

**Figure 6 nutrients-13-02665-f006:**
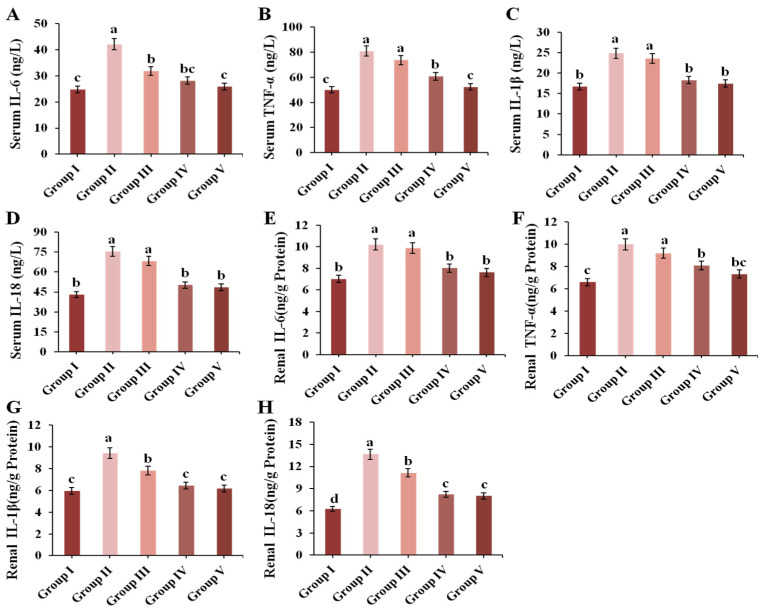
Effect of CPE-E on serum IL-6 (**A**), TNF-α (**B**), IL-1β (**C**), IL-18 (**D**) and renal IL-6 (**E**), TNF-α (**F**), IL-1β (**G**), IL-18 (**H**) in hyperuricemic mice. Group I was the control group. Group II was given potassium oxonate. Group III and Group IV were given potassium oxonate plus 2 and 4 g/kg BW of CPE-E, respectively. Group V was given potassium oxonate plus 5 mg/kg BW of allopurinol. Different lower case letters correspond to significant differences at *p* < 0.05.

**Figure 7 nutrients-13-02665-f007:**
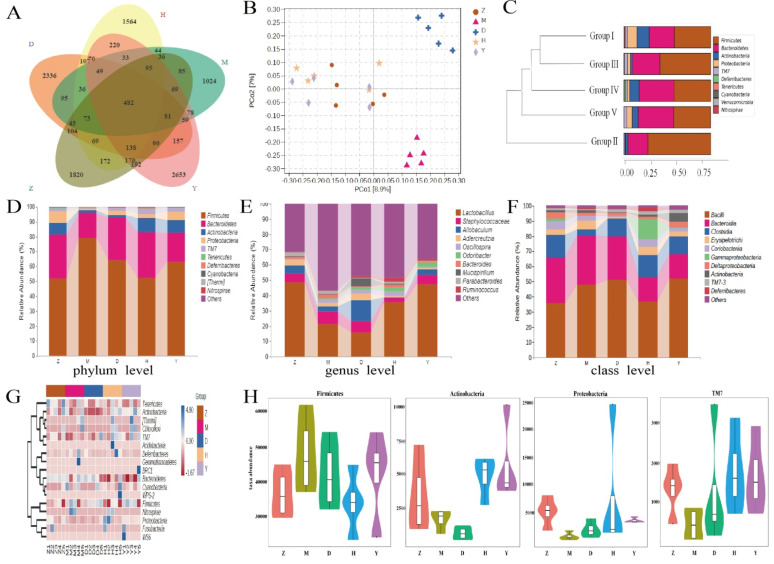
The effect of CPE-E on gut microbiota structure. Venn diagram of OUT (**A**), PCoA (**B**), hierarchical clustering (**C**), abundance distribution at phylum level (**D**), abundance distribution at genus level (**E**), abundance distribution at class level (**F**), Heatmap (**G**), absolute abundance of gut microbial community at phylum level (**H**). Z: Group I (control group), M: Group II (potassium oxonate), D: Group III (2 g/kg BW CPE-E), H: Group IV (4 g/kg BW CPE-E), Y: Group V (5 mg/kg BW allopurinol).

**Figure 8 nutrients-13-02665-f008:**
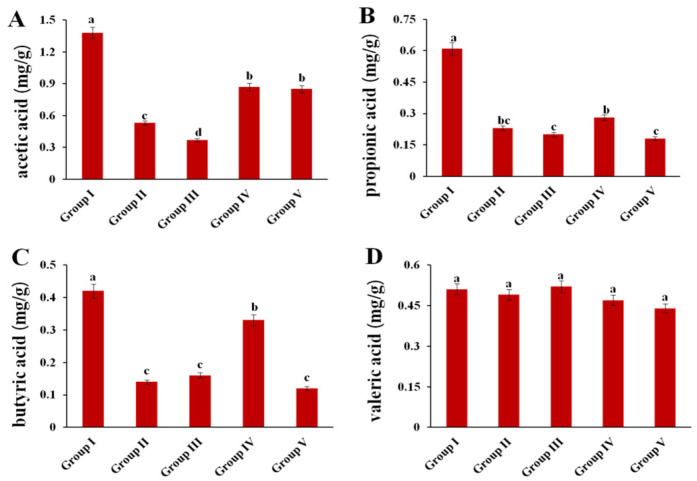
Effect of CPE-E on SCFA in hyperuricemic mice. Acetic acid (**A**), propionic acid (**B**), butyric acid (**C**) and valeric acid (**D**). Group I (control group), Group II (potassium oxonate), Group III (2 g/kg BW CPE-E), Group IV (4 g/kg BW CPE-E), Group V (5 mg/kg BW allopurinol). Different lower case letters correspond to significant differences at *p* < 0.05.

**Table 1 nutrients-13-02665-t001:** Identity and quantify of individual phenolics in CPE.

Peak No.	Retention Time (min)	Phenolic Compound	Content (mg/g)	
			CPE-E	CPE-H
1	4.21	gallic acid	5.03 ± 0.21	nd
5	19.64	*p*-hydroxybenzoic acid	1.28 ± 0.13	1.54 ± 0.09
6	21.39	ferulic acid	1.90 ± 0.15	2.40 ± 0.13
8	29.20	ellagic acid	2.26 ± 0.12	4.35 ± 0.21
10	36.98	quercetin	6.85 ± 0.23	nd
12	40.81	kaempferol	7.29 ± 0.19	nd

Note: “nd” means not detected.

**Table 2 nutrients-13-02665-t002:** Antioxidant activities of CPE in vitro.

Samples	TPC (mg GAE/g)	TFC (mg RE/g)	FRAP (mg Trolox/mg)	DPPH (IC_50_ mg/mL)	Chelating Activity (mg Na_2_EDTA/g)
CPE-E	136.63 ± 4.32 ^a^	67.49 ± 4.64 ^a^	0.49 ± 0.01 ^a^	0.20 ± 0.02 ^a^	29.10 ± 8.49 ^a^
CPE-H	31.74 ± 0.21 ^b^	15.26 ± 0.38 ^b^	0.07 ± 0.01 ^b^	5.03 ± 0.03 ^b^	12.93 ± 1.03 ^b^

Note: The data after “±” is the standard deviation (*n* = 3); ^a,b^: different lowercase letters in the same column of data represent significant differences (*p* < 0.05).

**Table 3 nutrients-13-02665-t003:** Microbial diversity index.

Groups	Chao1	Shannon	Simpson	Observed Species
Group I	1426.14 ± 165.01 ^a^	6.89 ± 0.54 ^a^	0.97 ± 0.015 ^a^	1429.87 ± 164.75 ^a^
Group II	1333.45 ± 175.10 ^a^	6.08 ± 0.39 ^b^	0.90 ± 0.039 ^b^	1322.59 ± 189.73 ^a^
Group III	1491.26 ± 303.42 ^a^	6.79 ± 0.59 ^a^	0.95 ± 0.018 ^a^	1402.30 ± 220.19 ^a^
Group IV	1508.36 ± 234.15 ^a^	6.95 ± 0.43 ^a^	0.98 ± 0.013 ^a^	1494.26 ± 303.43 ^a^
Group V	1463.56 ± 275.42 ^a^	6.91 ± 0.78 ^a^	0.96 ± 0.038 ^a^	1470.93 ± 245.26 ^a^

Note: Group I was the control group. Group II was given potassium oxonate. Group III and Group IV were given potassium oxonate plus 2 and 4 g/kg BW of CPE-E, respectively. Group V was given potassium oxonate plus 5 mg/kg BW of allopurinol. Different lower case letters correspond to significant differences at *p* < 0.05.

## Data Availability

The data presented in this study are available on request from the corresponding author.

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
