# Peer review of "Impact of Camellia japonica Bee Pollen Polyphenols on Hyperuricemia and Gut Microbiota in Potassium Oxonate-Induced Mice"

_nutrients, 2021, doi:10.3390/nu13082665_

Round 1

Reviewer 1 Report

Nutrients-1303592 Manuscript review

This manuscript describes the effects of a bee pollen extract (CPE) in the context of hyperuricemia. The authors include analytical assessment of the extract and a number of antioxidant assays in vitro, providing adequate redundancy in that regard. In addition, a mouse model of hyperuricemia was used to evaluate the in vivo effects of CPE, using a positive control (AP) and several redundant assays, spanning inflammatory cytokines, inflammatory pathways via gene expression, and the potential involvement of the gut microbiota was also explored. Results are rather abundant and well explained; however, several improvements can be made, especially to the discussion section, as per the comments below:

Abstract:

  • Please rephrase the aims of the study as they are not clearly stated
  • The term “super protection” is vague; I recommend removing it

Introduction:

  • Please provide more information regarding possible mechanisms by which the gut microbiota participates in purine/uric acid metabolism, and more information regarding the nature of the relationship between the gut microbiota and hyperuricemia, in terms of the possible direction of causality.

Methods:

  • Line 150: is it 15 µL or mL?
  • Line 174: what type of tube was used for serum collection and what was the centrifugation speed?
  • Histopathological examination: was the person performing the microscopic examination blinded? Did more than one person perform these analyses?
  • Line 204: Please describe how the microbial DNA was extracted.
  • Statistical analysis: Only the post-hoc test (Tukey’s test) is mentioned, please specify if the effect of treatments was assessed using one-way ANOVA.

Results:

  • Table 1: how many (technical) replicates are represented by this data?
  • Line 234: TPC, TFC and antioxidant activity are shown in Table 2, not Table 1
  • Table 2: statistical tests were performed to compare the 2 extracts, please specify in the methods section if it is a t-test or other test.
  • Figure 3: Panel E contains characters not in English
  • 5: Please refer to “renal tissue” rather than “renal”
  • Figure 4: please specify the unit used for mRNA levels
  • Figure 5: please indicate the scale on the images
  • Line 371: the definition of alpha-diversity here is incorrect
  • Line 374: It is stated that the PO-induced model group had reduced Chao1 and Observed species, however the table shows that these reductions were not statistically significant
  • Line 377: What do you mean by “the increase of species richness and evenness was explained to certain extent”? Explained by what?
  • Figure 7B,D,E,F,G: The legend is not clear as to which symbol represents which treatment; please use similar designations to other figures for clarity. The legend for panel H is not visible, so the reader is unable to interpret the figure.

Discussion:

  • Line 442: do you mean uricase (not urease)
  • The discussion is rather repetitive and reads like a more detailed repetition of the results. More discussion points should be included, especially in relation to previous literature, the strengths and limitations of these studies.
  • The involvement of the TLR4/MyD88/NF-kB pathway and other pathways should be further discussed in the context of hyperuricemia.
  • A further discussion of possible mechanisms of action of CPE would strengthen the discussion.
  • A discussion of the potential application of CPE in the clinical context should be included.
  • A discussion regarding the fact that mice were pre-treated with CPE before induction of hyperuricemia should be included. What indications are there of the possible efficacy of administering CPE in patients with established hyperuricemia?

Other:

The manuscript also requires extensive English syntax and grammar editing, for example:

  • The word “hyperuricemic” is used throughout the manuscript instead of “hyperuricemia” to describe the condition, please correct.
  • Line 10: “… is one of the major types of bee pollen in China” rather than “one of the major bee pollen”
  • Do not start a sentence with a numerical character (use three instead of 3 for example).
  • Make sure to use only the past tense for all verbs in the methods section.
  • Make sure to use complete sentences (for example: “Thirty µL were added to the solution” rather than “Added 30 uL… “).
  • “Inflammatory cytokines” rather than “inflammation cytokines”
  • “signaling pathways” rather than “signal pathways”

Reviewer 2 Report

Authors investigated the effects of two doses of Camellia Japonica bee pollen polyphenol extract on hyperuricemia and gut microbiota in potassium oxonate-induced mice. The manuscript has some good information for readers and the scientific community. The gut microbiome result is still preliminary but looks promising for further investigation. However, this reviewer has some comments/concerns and recommended major revision and re-submission.

  • The language and grammar in the present manuscript require major improvement. Please check carefully to eliminate grammatical errors.
  • The description of animal trial is not clear. What is the sample size in each group? Which group received potassium oxonate treatment? How were the mice treated?
  • Statistical analysis requires major improvement.

Specific points:

Title:

The disease is called hyperuricemia. Change hyperuricemic to hyperuricemia. Please check through the manuscript.

Abstract:

Line 13: The use of “resulting in” may not be proper. The causal relationship between CPE-E suppressed activation of several pathways and the reductions of inflammatory cytokines and the modulation on gut microbiota structure was not investigated.

The result of gut microbiota was not mentioned in the abstract.

Introduction:

“Gut microbiota participates in metabolism of purine and uric acid. Hyperuricemic affected the composition and metabolism (short chain fatty acid, SCFA) of gut microbiota.”

Please provide the reference for these statement.

What is XOD? Please present the spelled-out version when first time you use an abbreviation.

“Here, we firstly report that Camellia Japonica bee pollen polyphenols can prevent hyperuricemic and renal injury.”

It is hard to draw this conclusion based on an animal study.

Method:

What is the mice number in each group? Is 10 the total mice number or mice number in each group?

What is the difference between normal group and model group? Was the model group given potassium oxonate? Was there any other group given potassium oxonate? What is the dosage? Please clarify these in the animal experiment section.

How is the dosage 2, 4 g CPE-E/kg BW equivalent to human dosage?

Which MiSeq reagent was used for sequencing? How many samples were pooled in a single run?

The description of bioinformatic analysis could be improved. What method was used to check the quality of raw sequences? Were low-quality reads trimmed?

SCFA analysis: Were the fecal sample in fresh weight or freeze dried? How did normalize the water content in fecal samples?

Please provide more details in statistical analysis. How was the sample size estimated? Tukey is just a post hoc test. Which test/model in SAS was used for statistical analysis? Were the data checked for normal distribution? Was the same statistical method used for all endpoints? What is the statistical method used for gut microbiota results? PCoA only presents the clusters. It is recommended to add PERMANOVA test results.

Results:

Figure 7F: Since the genus result was shown in Figure 7E, it is recommended to present the species level data instead of the class level.

Conclusions:

“In conclusion, CPE-E, possessing abundant polyphenols and high antioxidant capabilities, could cut down production of uric acid and promote excretion of uric acid.”

It is hard to conclude this statement according to the results.

Round 2

Reviewer 2 Report

The author addressed some of my questions. However, there are still a few questions/comments not addressed.

Is the MiSeq Reagent Kit V3 600 cycles or 150 cycles? How many samples were pooled in a single run?

Please clarify in the method section that freeze-dried fecal samples were used for SCFA analysis.

For the statistical analysis:

Were the data checked for normal distribution? What did you do if the data were not normal distributed? Normal distribution is one of the assumptions of ANOVA. If data fails normal distribution, ANOVA is invalid.

How was the sample size estimated? Will current sample size provide enough power for your primary outcome? Considering the large inter-individual variation of gut microbiome, is 5 samples enough to observe the treatment effect?

For the statistical method used for gut microbiota results, please provide more details. For example, which statistical method was used to compare the beta-diversity between the groups? Which method was used to identify the genus/class that significantly changed?  PERMANOVA test is available at QIIME.
